# Mothers with Cancer: An Intersectional Mixed-Methods Study Investigating Role Demands and Perceived Coping Abilities

**DOI:** 10.3390/cancers15061915

**Published:** 2023-03-22

**Authors:** Athina Spiropoulos, Julie Deleemans, Sara Beattie, Linda E. Carlson

**Affiliations:** 1Division of Psychosocial Oncology, Department of Oncology, Cumming School of Medicine, University of Calgary, Calgary, AB T2N 4N1, Canada; 2Tom Baker Cancer Centre, Alberta Health Services, Calgary, AB T2N 4N2, Canada

**Keywords:** mothers, cancer, role demands, coping, intersectionality

## Abstract

**Simple Summary:**

Mothers with cancer feel shame and guilt when they struggle to balance their roles as parents and patients. There is a lack of research on how mothers with cancer cope with multiple role demands, and how their identities as women and persons experiencing a disability influence their coping strategies. This study investigates the roles that mothers with cancer assume and their perceived coping ability using an intersectional framework. The Role Coping as a Mother with Cancer (RCMC) model was created to demonstrate the multiple roles that mothers with cancer occupy, how they emotionally and practically cope with role conflict, and the identities that inform their experiences. This is a critical step in understanding the multifaceted nature of motherhood and how it is influenced by the demands of cancer treatment, providing an intersectional model to help inform further integrative research and clinical care.

**Abstract:**

Mothers with cancer report guilt associated with failing to successfully balance their parental roles and cancer. This study utilized a cross-sectional mixed-methods design and intersectional framework to investigate the multiple roles that mothers with cancer assume and their perceived coping ability. Participants included mothers diagnosed with any type or stage of cancer, in treatment or ≤3 years post-treatment, and experiencing cancer-related disability with a dependent child (<18 years, living at home). Participants completed a questionnaire battery, semi-structured interview, and optional focus group. Descriptive statistics, correlations, and thematic inductive analyses are reported. The participants’ (*N* = 18) mean age was 45 years (*SD* = 5.50), and 67% were in active treatment. Their role participation (*M* = 42.74, ±6.21), role satisfaction (*M* = 43.32, ±5.61), and self-efficacy (*M* = 43.34, ±5.62) were lower than the general population score of 50. Greater role participation and higher role satisfaction were positively correlated (*r* = 0.74, *p* ≤ 0.001). A qualitative analysis revealed that the mothers retained most roles, and that their quality of life depended on their capacity to balance those roles through emotion-focused and problem-focused coping. We developed the intersectional Role Coping as a Mother with Cancer (RCMC) model, which has potential research and clinical utility.

## 1. Introduction

In general, mothers occupy many social roles and are expected to take on the majority of parental and household demands, regardless of the other roles they occupy [1]. When role demands (i.e., the responsibilities associated with the roles an individual occupies) clash, an individual may experience role strain, which is characterized by the stress experienced when an individual cannot meet the demands of their social roles [2].

### 1.1. Role Strain due to Cancer-Related Disability

In 2013, global estimates of the years lost to disability indicated that cervical and breast cancer contributed to the highest number of women living with disability [3], and 96% of women with breast cancer experienced cancer-related disability [4]. It is common for cancer patients and survivors to struggle with cancer-related fatigue [5,6,7] and cognitive impairment [8,9,10]. Symptoms can persist for years after the end of treatment [11].

Mothers with cancer express feeling morally obligated to overcome cancer but simultaneously prioritize the needs of their children [12,13,14]. Guilt surrounding a perceived failure to meet parenting expectations may be accompanied by lower self-efficacy, higher levels of stress, and higher levels of anxiety, even among mothers who do not buy into the “good mother” ideal [15]. Moreover, mothers report experiencing distressing role conflicts surrounding cancer [16], employment [17,18], romantic relationships [19], housework [20], and self-care or leisure activities [18]. Understanding the interactions between the multiple social roles that mothers with cancer occupy may provide a more comprehensive understanding of this population and opportunities for improved care.

### 1.2. The Importance of Occupying Multiple Social Roles

Practical constraints may necessitate participation in certain social roles, such as employment, as cancer patients are at an increased risk for financial strain [21]. Marital status can impact role demands such that single mothers with cancer may assume increased role demands to support children [22]. Despite the potential for added stress, previous research on role expansion theory suggests that participation in multiple roles may have some beneficial effects, such as increased social support, self-complexity, opportunities to experience success [23], and a decreased risk for long-term health challenges (e.g., insomnia, lingering illness, medication dependency) [24].

Understanding the sociocultural context surrounding mothers is critical, as the authors of [25] write: “women’s health involves their emotional, social, cultural, spiritual, and physical well-being. It is determined not only by biology, but also by the social, political, and economic context of their lives” (p. 1047). Implementing effective coping strategies may be a critical component for mothers’ mental health within their multiple roles, and the coping strategies accessed may be informed by sociocultural expectations [26]. People generally utilize emotion-focused coping strategies (i.e., actively identifying, processing, and expressing one’s emotions) [27] and/or problem-focused coping strategies (i.e., measures taken to solve or reduce the stressor directly by modifying the person–environment relationship) [28]. However, the ways in which mothers utilize these strategies to manage their multiple role demands throughout cancer treatment requires further investigation, as it may be critical in supporting mothers in participating in their desired roles.

### 1.3. Applying an Intersectional Framework

Intersectionality theory considers an individual’s lived experience in the context of the multiple identities they occupy and how the sociocultural assumptions surrounding these identities interact [29]. This is important as sexism and ableism exist within healthcare for women [30,31].

Socialist feminist theory connects women’s positionality within society to the ways in which women, their labor (formal and informal), and their needs are devalued due to the patriarchal system they operate within [32,33]. The pressure for mothers with cancer to meet and prioritize parental labor can be exacerbated by culturally salient gender role norms that dictate that mothers should prioritize family over self and serve as caregivers, not as dependents [34].

Critical disability theory (CDT) contests hegemonic understandings of disability, including the tendency to pathologize, medicalize, and essentialize impairments [35]. The supercrip schema suggests that society determines disabled people’s value based on their ability to overcome their disability and achieve “normalcy” [36]. CDT and socialist feminist theory, when taken together, may help fill the knowledge gap regarding how social perceptions surrounding womanhood and disability impact mothers with cancer. The present study was designed using an intersectional framework to capture the sociocultural context that influences mothers’ personal and healthcare relationship dynamics and the strategies they use to cope with their multiple role demands.

### 1.4. Primary Study Aims

The present study utilizes a critical inquiry paradigm and methodology grounded in intersectionality theory to examine how mothers with cancer cope with their multiple role demands. For mothers with cancer, the study aims to (i) determine what roles mothers assume and how cancer impacts role functioning; (ii) identify how mothers’ perceived ability to cope with role demands impacts psychosocial functioning and quality of life (QoL); (iii) explore how the sociocultural context surrounding disability and womanhood impacts role functioning after a cancer diagnosis.

## 2. Materials and Methods

### 2.1. Study Design

The present study was a quant + QUAL additional-coverage concurrent mixed-methods design [37] that included a questionnaire battery, interview, and optional focus group session. A mixed-methods approach was appropriate given its usefulness at exploring complex sociocultural dynamics and personal perceptions [38], both of which the present study aimed to address. Participants completed one 20 min questionnaire battery, which collected demographic, clinical, and health data; it also included National Institutes of Health Patient-Reported Outcomes Measurement Information System (PROMIS) measures, which are reliable and validated patient-reported outcome tools [39]. After completing the questionnaire battery, participants completed one semi-structured interview ranging from 45 to 120 min over Zoom or phone, depending on the participant preference. Once the qualitative and quantitative data analyses were completed, participants were invited to participate in one of two optional 90 min focus group sessions over Zoom centered around the participant evaluation of the research findings.

### 2.2. Participants

Eighteen participants were recruited from across Canada using convenience and snowball sampling through online advertisements, and from the Tom Baker Cancer Center (TBCC), Wellspring, Alberta, and social media targeted at young adult cancer groups. Mothers were given additional study information and were screened for eligibility by a member of the study team. Inclusion criteria consisted of women with cancer who were living in Canada, diagnosed with any type or stage of cancer, in active treatment or ≤3 years post-treatment, with a dependent child (<18 years, living at home), and currently experiencing cancer-related disability according to the Statistics Canada-derived RACDPAL variable in the Canadian Community Health Survey [40]. The exclusion criterion included patients with a significant cognitive or developmental disability prior to a cancer diagnosis. Participants were deidentified and given pseudonyms to be used in all published material associated with the present study.

### 2.3. Measures

#### 2.3.1. Questionnaire Battery

Demographic (e.g., age, annual income, number of dependent children, etc.), clinical (e.g., time since diagnosis, cancer type, treatment type, etc.), and health (e.g., exercise frequency, alcohol use, etc.) data were collected. Participants were given the option to skip any demographic, clinical, or health questions that they were not comfortable answering. Psychological functioning was evaluated using the PROMIS—Cancer Item Bank v1.0—Emotional Distress—Anxiety Short Form 6a questionnaire, and the PROMIS—Cancer Item Bank v1.0—Emotional Distress—Depression Short Form 6a questionnaire. Role functioning was evaluated using the PROMIS—Item Bank v2.0—Ability to Participate in Social Roles and Activities Calibrated Items questionnaire, the PROMIS—Item Bank v1.0—Satisfaction with Social Roles Calibrated questionnaire, and the PROMIS Item Bank v1.0—Self-Efficacy for Managing Chronic Conditions—Managing Social Interactions questionnaire.

#### 2.3.2. Semi-Structured Interviews and Focus Groups

The semi-structured interviews followed an interview guide, which was collaboratively developed by members of the study team based on previous literature to explore the three study aims (see Appendix A). All interviews were conducted by the first author. Focus groups followed a session handout, which was collaboratively developed by members of the study team to perform a participant authenticity check and provide recommendations regarding knowledge translation (see Appendix A). All focus groups were co-facilitated by two members of the study team (A.S. and S.B.).

### 2.4. Data Analysis

Aim (ii) was assessed by converting raw scores for each PROMIS measure into t-scores using the HealthMeasures Scoring Service (http://www.healthmeasures.net accessed January 2022), and by conducting statistical analysis using IBM SPSS Statistics 26.0, with the alpha set at *p* = 0.05. Due to the small sample size, the quantitative analysis was descriptive and exploratory, as this analysis is not adequately powered to make generalizable claims. Aims (i), (ii), and (iii) were assessed by recording and transcribing the semi-structured interviews verbatim and coding them using NVivo 12 [41]. The quantitative and qualitative results were integrated to form overall conclusions. According to the method in [42] for assessing thematic saturation in qualitative research, and using a base size of 4, we reached the ≤5% new information threshold at 4^+2^ interviews. Thus, our sample size was sufficient. The focus groups were also recorded and transcribed verbatim and were coded using NVivo 12 [41], following which the themes and models derived from the interviews were adjusted to reflect the participant feedback.

#### 2.4.1. Questionnaire Battery

The demographic, clinical, and health characteristics of the participants were described using descriptive statistics and frequency analysis. The PROMIS t-scores were aggregated and compared to the respective PROMIS reference populations (*M* = 50, *SD* = 10) following the scoring guidelines. The relationships between psychological functioning (anxiety, depression), role functioning (role participation, role satisfaction, self-efficacy at managing social roles), and the demographic characteristics that may relate to the number of role demands placed on a mother (e.g., income, number of dependent children, marital status, etc.) were determined using Pearson product–moment correlations.

#### 2.4.2. Semi-Structured Interviews and Focus Groups

The semi-structured interviews and focus group sessions were analyzed using a thematic inductive approach (see Table 1 and Appendix A, respectively) [43]. The credibility of the semi-structured interviews was addressed using frequent peer (consensus approach between three coders) and participant review authenticity checks (focus groups).

## 3. Results

### 3.1. Questionnaire Battery

#### 3.1.1. Demographic, Clinical, and Health Characteristics

Eighteen participants completed the questionnaire battery, with no dropouts. The participants’ demographic, clinical, and health characteristics are shown in the Appendix A (Table 2). All screened mothers qualified as experiencing a disability, and 83% of the participants reported that cancer-related disability reduced the amount or kind of activities they could perform daily. A total of 89% of the participants reported that cancer-related disability reduced the amount and/or kind of activity they could perform at home, and 91% of the participants who were working during treatment indicated that cancer-related disability impaired the amount and/or kind of activity they could perform at work. The participants’ mean age was 45 years (*SD* = 5.5), ranging from 35 to 53 years (22% in their 30s, 56% in their 40s, 22% in their 50s), and they were predominantly married (67%), Caucasian (75%), middle-class (56%, >120,000 annually), and mothers of one child (55%). Most participants were currently in treatment (67%), had breast cancer (72%), and had undergone two or more types of cancer treatment (78%). Among the participants who had completed their cancer treatments, the mean time off treatment was 8.8 months.

#### 3.1.2. Psychological and Role Functioning Compared with General Population

The mothers reported greater levels of anxiety (*M* = 54.90, *SD* = 8.22) and depression (*M* = 54.88, *SD* = 4.69) compared with the general healthy reference population (*M* = 50). In the cancer population, these scores indicate mild symptom severity [44]. The mothers reported poorer role participation (*M* = 42.74, *SD* = 6.21), role satisfaction (*M* = 43.32, *SD* = 5.61), and self-efficacy at managing social roles (*M* = 43.34, *SD* = 5.62) compared with the general healthy reference population (*M* = 50).

#### 3.1.3. Associations between Psychological and Role Functioning

The associations between psychological and role functioning are shown in Figure 1.

The correlation analysis revealed that increased role satisfaction was positively correlated with greater role participation (*r* = 0.74, *p* < 0.001) and negatively correlated with lower depression (*r* = −0.64, *p* < 0.004). The more a mother felt she was able to participate in her roles, the more satisfaction and less depressive symptoms she experienced.

There were no significant correlations between anxiety and any of the domains of role functioning. Moreover, no significant associations between any of the domains of role functioning and demographic characteristics, such as household income, employment status, relationship status, number of dependent children, or age of dependent children, were detected. However, an exploratory analysis found that the frequency of participating in hobbies was negatively correlated with anxiety (*r* = −0.607, *p* = 0.008) and depression (*r* = −0.666, *p* = 0.003). Thus, the more the mothers participated in hobbies, the lower the anxiety and depression they experienced. Given the sample size, these results are purely exploratory and should be interpreted with caution.

### 3.2. Semi-Structured Interviews

A qualitative analysis of 18 semi-structured interviews led to the emergence of 4 themes: (1) navigating roles and role conflict, (2) changing self-concept, (3) strategies used to cope with role demands, and (4) perceived QoL.

#### 3.2.1. Theme 1: Navigating Roles and Role Conflict

This theme was discussed the most and four subthemes emerged: parental, cancer, relationship, and work role demands (Figure 2).

Overall, the mothers continued to occupy most if not all of their social roles prior to cancer diagnosis. Their emotional labor demands and organizational responsibilities usually increased within their parental, relationship, and household roles:

“*I’m a mom, I’m a girlfriend, sister, friend, an employee, the housewife, I clean, I cook, and above all the things, an independent woman.*”(Melanie)

*Parental Role Demands.* This was the most frequently discussed role, and all the participants identified that it was the most prioritized during cancer treatment. The mothers’ parental role demands differed depending on their children’s developmental stage, pre-existing relationships, and the mother’s cancer prognosis. The mothers of toddlers recalled that their children remained highly physically dependent but reported that their toddlers were oblivious to their cancer diagnoses and often a helpful reprieve:

“*She was just like a happy and joyful little [toddler] and I think we all, like all of us, her brothers, my husband, and grandparents and all that like really got in escape like in playing with her joyfulness.*”(Kristina)

Elementary-aged children largely understood cancer relative to how it affected their lives, and this intensified the pressure the mothers felt to meet parental role demands. The mothers reported that their children showed simplistic signs of being distraught or worried, and that this was often spurred on by physical manifestations of cancer, such as hair loss. Older elementary-aged children still struggled to comprehend their mothers’ cancer but were beginning to think more deeply and understood the concept of mortality and how it might affect their lives:

“*Even at [older-elementary age] developmentally they’re starting to understand a few more things about the world.—And mortality, right. She had a fish that died. She understands that there’s other people—even in COVID that there’s other people that, do you know what I mean? Mortality is a bit of a real thing and I think that there’s been, she’s starting to kind of get a little bit of grasp with that, and so there are certain levels of anxiety that come out just from the sheer understanding that there’s a big universe out there and that people are mortal.*”(Lauren)

Children in early adolescence were generally characterized as highly self-centered, and their desire for increased independence and control made it challenging for the mothers to delegate additional demands to them. Moreover, early adolescents were unforgiving when cancer-related role demands impacted the expectations they had of their mothers, and they were more likely to externalize their stress in maladaptive ways:

“*He is going through the teenage crazies basically. I don’t even know. He is smoking weed; he is sneaking out. Like all the f **** things. I never would have suspected he would be a kid like that, but that is what is happening and it’s just like all happening around the same time as this cancer stuff, which is honestly a million times more stressful.*”(Alex)

Children in mid-to-late adolescence were able to conceptualize their mothers’ cancer diagnoses and subsequent limitations in a nuanced way, support their mothers by taking on increased role demands, and advocate for their own emotional support needs.

The parent–child dynamic, and how the mothers navigated their role demands, were highly variable across cancer prognoses. For mothers with early-stage or nonaggressive cancers, changes to their ability to meet their parental role demands were viewed as temporary and manageable. Conversely, mothers with poor or terminal prognoses struggled with losing the opportunity to watch their children grow and expressed concern about how the physical and emotional parental role demands they occupied would be redistributed once they passed away:

“*You start thinking about your own mortality and you imagine that how much time might I have. I think a lot about how that’s going to impact my son’s life, my ability to be a participant in his life, how long am I going to be alive for him. That was the biggest difficulty. I know my husband would struggle, but he’d move on, but I always think about the impact on my son.*”(Gabriella)

The quantitative analysis revealed no relationship between the number or age of children and psychological or role functioning. Although the quantitative findings are exploratory, they are consistent with the overall qualitative finding of this theme that physical and emotional parental demands change inversely throughout development; while the challenges faced varied, the total amount of role conflict did not.

*Cancer Role Demands.* Cancer-related role demands were the second most prioritized and included managing treatment schedules and side effects, as well as the steps taken by the mothers to regain control over their bodies and roles. Treatment demands heavily influenced the time and energy that the mothers had to fulfill their other roles. Persistent cancer-related disability, particularly for the survivors who had hoped that they would return to normal once they completed cancer treatment, made the mothers feel as though their cancer-related role demands would always detract from their desired roles.

*Relationship Role Demands.* Overall, role demands associated with romantic relationships, friendships, and being caregivers to parents were the mothers’ third priority during cancer treatment, as they were viewed as the most elastic. All the partnered participants were in heterosexual romantic relationships, which were subject to considerable role conflict with parental, household, and cancer demands, and the degree to which the mothers were satisfied with how their relationships responded varied dramatically. Relationships with histories of successfully navigating role conflict were generally more resilient to the shifting role demands caused by cancer. Conversely, some mothers identified that their romantic relationships struggled to adapt to the pressures of cancer and the resulting changes to role demands. The expectations surrounding physical intimacy demands changed, and while most mothers desired non-sexual intimacy during treatment, their interest in sexual intimacy generally declined, resulting in decreased closeness:

“*When you go through chemos everything in your body is changing hormones, you’re exhausted. There was zero intimacy. I don’t think my husband and I did more than a kiss for probably a year.*”(Nicole)

How parental and household role demands were distributed was often linked with relationship satisfaction. Mothers in unstable romantic relationships felt uncomfortable asserting their needs to their partners, and most reported challenges surrounding their partners’ initiative and ability to complete tasks to their standards, which forced them to reclaim the tasks. A salient factor that impacted how the mothers prioritized and navigated role conflict was marital status. Regardless of the mothers’ satisfaction with their relationships, the mothers who were dating and not living with their partners were less likely to ask for support compared with those in long-term partnerships. Single mothers were particularly firm in terms of prioritizing their role as a parent, which caused conflict with partners who felt emotionally forgotten:

“*I just feel like I don’t have enough time to input into all these relationships that I need to. So, they’re always at odds with each other, my daughter and my boyfriend who don’t get along.*”(Lynn)

In addition to spousal relationship role demands, the mothers also navigated demands associated with their role as friends. Friendships were the most consistently reported as positive influences on the mothers’ lives. Some mothers chose to deprioritize their role as a friend, and others chose to invest more time in these relationships. However, in all cases, maintaining a network of friends was characterized as highly beneficial and one of the easiest roles for mothers to occupy. Mothers’ relationships with parents and in-laws often flipped in terms of caregiving and independence. Several mothers went to live with their parents and redistributed some household and parental role demands while undergoing treatment.

*Work Role Demands.* Mothers’ work-related role demands were the lowest priority during cancer treatment. Work demands included informal work (household tasks) and formal work (paid and unpaid employment). All mothers, however, experienced informal work demands, such as maintaining a home, which were de-emphasized despite the mothers largely retaining these responsibilities. Employed mothers either sought disability support or cut back to part-time work, and they were generally satisfied with the extent to which their employers and coworkers accommodated them. Eight mothers were grateful to relinquish the demands of formal work, but five mothers reported feeling deeply connected to their work and that having to take disability leave was highly distressing. The mothers reported varying types of work, their work satisfaction prior to receiving their cancer diagnoses, and the financial need to continue working after their cancer diagnoses; these may be modifying/qualifying factors in the relationship between mothers’ employment status and psychological and role functioning. Concerns surrounding returning to work, and the resulting impact on other role demands, were pressing issues, as cancer-related disability was still impacting functioning after the mothers completed their primary cancer treatments. In the interviews, the mothers were split between full-time work, part-time work, and homemaking, and they wanted to focus on the other roles they occupied. It is unclear whether data saturation was reached regarding the mothers who were managing formal work role demands. Hence, the conclusions drawn regarding this theme must be interpreted with caution.

“*I think that there was an expectation that mom would look after the food and mom would look after cleaning and mom would—so mom would take care of the home.*”(Gabriella)

#### 3.2.2. Theme 2: Changing Self-Concept

This theme was the least frequently discussed and three subthemes emerged: changes to a mother’s caregiving capacity, sense of autonomy, and relationship with femininity after diagnosis (Figure 3).

*Caregiving Capacity.* The mothers’ self-concept was impacted by their capacity to care for family and friends physically and emotionally; this changed during cancer treatment due to role separation, high personal standards for role participation, and the perception that they were now a burden. Switching from a caregiver to dependent role made the mothers feel separated from the role as they struggled to redefine themselves within a new relationship dynamic. Fifteen mothers reported concerns regarding being a burden if they were unable to meet all their role demands, which made this the most frequently discussed concern regarding struggling to manage role conflict:

“*I feel pretty guilty because I’ve always prided myself on being really reliable and dependable, and I’m not that way anymore.*”(Monica)

The mothers generally did not feel that family or friends had unfair expectations, and they characterized them as “understanding” and “accepting.” The mothers reported that they were internally motivated to take on role demands at the expense of their mental health and presented an incongruent ideal self.

*Notions of Autonomy.* Prior to their cancer diagnoses, the mothers felt relatively in control of their lives; however, this understanding of themselves shifted when cancer-related disability impacted their bodily autonomy and dependency. Notions of autonomy were reported as the second most frequent as it relates to changing self-concept. Losing bodily connection and autonomy due to cancer-related disability was a challenge for the mothers. The presence of cancer was often perceived as the body turning against itself, particularly when mothers faced cancer-related disability that interfered with their capacity to meet their role demands. Although the mothers knew that they were experiencing treatment side effects, many still found it hard to remove their perceived physical, emotional, or intellectual capabilities from their present experiences. Cancer treatment and the subsequent cancer-related disability made the mothers aware of their dependency on those around them:

“*It’s very frustrating for me not to be able to do things on my own, especially after my surgery—it’s detrimental to my mental health not to be able to be independent and I liked that, I liked that part of my identity. I think I worked really hard to get there and to be self-sufficient and learning skills to be able to do things on my own and to lose that ability has been very difficult.*”(Lynn)

Retaining a sense of independence was important to the mothers, and particularly to the single parents, as their identities were more closely tied to self-sufficiency. Navigating independence became increasingly unclear as the mothers completed their primary cancer treatments; they reported being socially encouraged to return to their roles in the same way they had before cancer, which was often not desirable or possible, and particularly for mothers on chronic maintenance therapies or with high chances of cancer reoccurrence.

*Relationship with Femininity.* For some of the mothers, cancer and cancer treatment interacted with their feminine identity following physiological changes and fertility challenges. The mothers discussed a range of physiological changes that resulted from cancer treatment, including mastectomies, drug-induced menopause, weight fluctuations, and hair loss. These changes pushed the mothers to consider how their physicality related to their sense of femininity. The mothers did not necessarily report that typically feminine characteristics, such as long hair or breasts, were required to be feminine, but rather that the process of watching these attributes change challenged their connection with womanhood.

Forced menopause was reported to be a particularly distressing treatment side effect for young mothers. Four mothers had to terminate their pregnancies, and others became infertile due to cancer treatment, which were among the most challenging forms of cancer-related disability to cope with:

“*She told me that because I asked about the pregnancy and she said because I was so early in my pregnancy if I did the chemo with the pregnancy it would cause the baby harm and pain and they would be deformed.—So, I had to decide if I was going to delay treatment and then hope and pray that my cancer—to give the baby a fighting chance or if I just didn’t have any treatment at all I deliver that baby to term and then that was the end of me or if I had to end the pregnancy that week to then start the chemotherapy.*”(Nicole)

#### 3.2.3. Theme 3: Strategies Used to Cope with Role Demands

The mothers with cancer were struggling to meet their role demands; however, all the participants had developed coping strategies to manage the consequences of role conflict, and this was the second most frequently discussed theme. Two subthemes emerged: emotion-focused coping strategies and problem-focused coping strategies (Figure 4).

All six mothers who had completed their primary cancer treatments ascertained that the coping strategies they used changed once they ended treatment, and that they struggled with the expectation of those around them that life return to normal in terms of the delegation of parental, relationship, and household role demands.

*Emotion-Focused Coping.* Emotion-focused coping strategies were reported slightly more frequently than problem-focused coping strategies and included accessing social support networks, practicing self-care, redefining personal expectations, and focusing on children. However, the mothers largely described deprioritizing emotion-focused coping strategies compared with problem-focused coping strategies until they completed their primary cancer treatments. Social support was viewed as the instrumental or emotional support provided by the members of a mother’s community (e.g., friends, family, coworkers), which may help to mitigate emotional or physical role strain. Relying on social support was the most reported strategy overall, it was used by all mothers, and it served a dual purpose for both problem-focused and emotion-focused coping:

“*I like to be surrounded by lots of people and I’ve actually made great new friends because of cancer for me and that’s like the best thing ever when I can expand my community.*”(Lana)

Prior to cancer, self-care was often overlooked by the mothers, who were preoccupied with managing their multiple roles. However, twelve mothers turned to self-care as a strategy to cope with cancer and the emotional burden of their role conflict, making it the second most utilized emotion-focused coping strategy. This supports the quantitative finding that the mothers who engaged in hobbies had significantly lower levels of anxiety and depression. Many of the mothers identified what form of self-care they needed by honoring what they physically required in the moment (e.g., napping, exercise, meditation, etc.):

“*Either meditate or take a nap and, and that’s what, that’s what my body needs in order to have the mental energy to, to think more clearly when I have more energy.*”(Anna)

The mothers noted that, while self-care significantly improved their wellbeing, engaging in these activities still occupied time that they could be spending on other role demands.

The mothers found it valuable to redefine their expectations within a role, and particularly when cancer-related disability required decreased role participation. This allowed the mothers to better cope with the stress, anxiety, and depression they struggled with when they were forced to withdraw from roles. When the mothers became disillusioned with their role functioning or the progression of their cancer treatment, and particularly for mothers managing terminal cancer, they often focused on their love and sense of responsibility towards their children.

*Problem-Focused Coping.* Problem-focused coping strategies were reported slightly less frequently than emotion-focused coping strategies and included continual communication, asserting needs to their family and broader support network, obtaining help from their partners, scheduling around cancer demands, and accessing professional services.

Communication was the most reported problem-focused coping strategy, and it was used, at least to some degree, by all the mothers. The extent to which cancer and treatment were discussed with children varied depending on their developmental stage. Communication was also a crucial strategy for couples struggling to maintain sexual intimacy. The mothers noted that they often refrained from sharing information with their friends and family until there was more certainty, but that when they did choose to share, it was important that the conversations were open:

“*It takes like extra, extra communication to be constantly talking about how our roles are going to change I guess on a daily basis even. What needs to be done and who can do what.*”(Kristina)

The mothers often relied on their social support networks to obtain instrumental help, such as driving children to school, delivering dinners, and researching cancer resources; in this way, social support plays a dual role in both emotion-focused and problem-focused coping. Despite the mothers noting that asking for and obtaining help to alleviate role demands was central to their ability to cope with their multiple roles and cancer, this was hard to do and the least frequently employed strategy. The extent to which the mothers felt that they could rely on their partners and children when they asserted their needs was highly variable and depended on the children’s age, the mothers’ marital stability prior to cancer, and their husbands’ employment demands:

“*It happened to be a time where my husband’s work was really busy, so he couldn’t really take more time off to do those things.*”(Irene)

Due to the relatively inflexible nature of cancer treatment, the mothers frequently reorganized their roles around their treatment schedules; this was the second most common strategy. Eight mothers accessed professional services. For example, meal delivery (e.g., Skip the Dishes), grocery delivery (e.g., Hello Fresh), and house cleaning services were utilized to relieve household role demands and prevent any conflict that could arise from delegating these demands to children and spouses. Professional counselling was utilized by three mothers who experienced strain on their romantic relationships because of their cancer diagnoses and subsequent role conflicts. However, this typically occurred once the mothers completed their primary cancer treatments and felt that they had the emotional capacity to delve into these issues. The mothers’ financial capacity impacted their ability to access professional services as a reliable coping strategy.

#### 3.2.4. Theme 4: Perceived QoL

Mothers with cancer have a unique and fluctuating relationship with their definition of QoL as it relates to their multiple role demands, and this was the third most frequently discussed theme. Two subthemes emerged: opportunities to re-evaluate life and seeking balance (Figure 5).

Overall, the mothers reported a satisfying QoL despite cancer-related disability and frequent role conflict, and their QoL related more to their psychosocial wellbeing and fulfillment than their physical state.

*Opportunities to Re-Evaluate Life.* Cancer served as a catalyst to re-evaluate what produced a personally fulfilling life, and this was the most frequently discussed contributing factor to QoL. The mothers reported shifting away from material pleasures to focus on experiences, the present, and building meaningful interpersonal connections. The prospect of death compelled the mothers to make immediate changes, but this shift in focus persisted for all the survivors:

“*I think the less you try to live by someone else’s standards and you really just try to be comfortable with your life,—everything you hear and see and what your girlfriends are telling you and what they post on Facebook it’s all noise and that you just have to find your rhythm. And the moment you can find your rhythm that’s where you find that peace.*”(Annette)

*Seeking Balance.* Achieving a sense of balance between the multiple roles a mother occupies was the second most frequently discussed contributing factor to QoL. The mothers described living a balanced life as a process of participating in desirable roles, having reciprocal relationships, and defining boundaries. What constitutes living a balanced life is highly personal, and the mothers reported that it was beneficial to try and separate their experiences from socially constructed notions of what balance looks like:

“*I think women have been given so many false representations of what balance looks like that’s not achievable.*”(Annette)

Although the mothers reported their relationships with family and friends to be generally beneficial, they also noted that these relationships needed to be reciprocal (i.e., meeting the demands and needs of the other person) to be fulfilling:

“*My friend got pregnant, she didn’t want to tell me because I wanted to have another baby and I might be infertile. And so they, but she didn’t want to tell me, right? So I feel like they’re scared to tell me their things, so I can’t even be [there] to support them with their stuff. So that’s really crappy, for sure.*”(Tracy)

The mothers did not want to retain all their role demands, but when multiple desired roles were flexible enough to accommodate their current needs, they were beneficial at fostering a sense of connection to their community, a sense of purpose, and cancer recovery overall. Being able to set boundaries regarding role participation was vital to achieving balance, and a precursor to defining boundaries was when the mothers allowed themselves to prioritize their own needs:

“*My work has been really good with letting me work two days a week. I find it gives me purpose, gets me out of bed, makes me feel normal to be honest with you, and that’s what I appreciate.*”(Bridgette)

Six mothers took their time on disability leave to consider how, or if, they wanted to re-enter the workforce in a way that aligned with their other roles. Moreover, the mothers reported wanting to set more boundaries in terms of their role demand distribution within the home. While these conversations became increasingly challenging once the mothers completed their primary cancer treatments, as the impact of the treatment or cancer-related disability became less apparent, they had an increased focus on achieving a balance between their roles by asserting their boundaries:

“*We needed to have that discussion. If this is still what should be done at home, then I am not going to work, because I can’t do it all.*”(Olivia)

#### 3.2.5. Proposing a New Model: Role Coping as a Mother with Cancer (RCMC)

The experiences and perspectives expressed by the mothers with cancer in the qualitative and quantitative data were combined into higher-order concepts to develop the Role Coping as a Mother with Cancer (RCMC) model (Figure 6). However, given that the small sample size limits quantitative validity, we prioritized qualitative results when developing the RCMC model. Building a model is valuable in exploratory research, as it comprehensively synthesizes lived experiences, which has clinical utility [45]. Unlike the visual maps, the RCMC model is designed to be an easily understood visual tool for patients and healthcare providers. All the dimensions of this model are in a continual and multidirectional state of influence and are divided into three components that integrate the research aims: (1) the interacting roles that mothers with cancer occupy (the size of the circle signifies the priority given to a role), (2) how mothers cope with their roles, and (3) the sociocultural context that mothers operate within. The interactions between roles can create, or alleviate, role conflict. Surrounding the interacting roles that a mother with cancer occupies is her capacity to cope with these roles according to qualitative and quantitative data. The outermost ring includes the identities that inform what it means to be a mother with cancer.

### 3.3. Focus Groups

A total of 2 focus groups were run, with 12 participants in total. A qualitative analysis of the focus groups led to the emergence of two themes: (1) participant authenticity check and (2) knowledge translation. The mothers largely agreed with the themes, but they wanted to emphasize how cancer impacted their relationship with womanhood and how that informed the way that they navigated their roles and coped with their role demands. Disagreements with the original themes were reflected in the themes presented. The mothers reported that the RCMC model accurately and comprehensively reflected their lived experiences, and that they were also satisfied with the visual construction of the model, indicating that the intertwined nature of their roles and the circular nature of coping effectively demonstrated their lived experiences. The mothers identified a need for the increased continuity of resource dissemination and an interest in seeing the RCMC model used in healthcare settings.

## 4. Discussion

### 4.1. The Roles Mothers with Cancer Assume and the Impact of Cancer

Overall, the mothers with cancer continued to occupy multiple intersecting roles, which caused role conflict. The mothers occupied four primary roles: parental, cancer, relationship, and work. They prioritized their role as a parent, followed by their role as a cancer patient, their role in relationships, and their role as an employee or homemaker. The mothers’ desire to prioritize their parental role demands is consistent with previous qualitative research on mothers with cancer [12,13,14,46]. The present study provides a novel understanding of the relationship between the multiple roles and how they are impacted by cancer in distinct yet intersecting ways. While some interactions create role conflict, such as cancer-related fatigue and the desire for sexual intimacy, other interactions help to alleviate role conflict, such as friends offering household or childcare support. Upon their cancer diagnoses, the mothers redefined QoL to center around a balance between their role demands in order to promote fulfilling role participation.

In a cross-sectional analysis of 6973 older cancer survivors, the amount of impairment that the participants experienced in their ability to complete daily living activities was moderated by the cancer type and cancer stage [47]. This is similar to the qualitative findings from the present study, as the extent to which the mothers experienced role conflict was modified by the treatment that they underwent to address their specific type and stage of cancer. For example, the mothers who were on maintenance chemotherapy had to permanently restructure their roles to accommodate the disability they experienced following treatment. However, those who only underwent one round of targeted therapy could sometimes redistribute their roles without having to consider the long-term impact on their sense of self and relationships.

### 4.2. Consequences of Maintaining Multiple Roles and how Mothers with Cancer Cope

The qualitative findings from the present study demonstrate that mothers want to participate in multiple roles during and after cancer treatment, and that they benefit from doing so. This finding provides evidence that suggests an alignment with role expansion theory, as in a longitudinal study of over eight years on 9000 randomly selected Swedes [24] that found that increasing the social roles that people participate in significantly improved health-related wellbeing. In contrast, in the present study, in terms of psychological wellbeing, quantitatively, there was only a significant negative correlation between role satisfaction and depression, and not role participation and depression, and no relationship with anxiety. This may be because the sample size was not large enough to find any association; however, because the qualitative analysis revealed a relationship between role participation and psychosocial wellbeing, this area warrants further exploration.

The exploratory quantitative findings from the present study describe a significant positive correlation between role satisfaction and role participation. Although this association should be interpreted with caution given the sample size, it is consistent with the qualitive findings and the literature. For example, in the present study, when the mothers felt as though they were participating in their roles, they were more satisfied with said roles, which is similar to previous research on the workplace that suggests that role participation mediates role satisfaction [48]. The present study provides evidence that role expansion theory should be considered when providing integrative cancer care for mothers with cancer, as mothers should be enabled to participate in multiple desired social roles throughout cancer treatment.

However, for the retainment of the desired roles during cancer treatment to be feasible and beneficial, mothers need to utilize emotion-focused and problem-focused coping strategies. A study of 80 women at risk for ovarian cancer assessed the relationship between perceived control, problem-focused coping, and psychological adjustment; the study found that, in cases in which women believed they were wholly in control of their health, problem-focused coping strategies resulted in increased distress, which was possibly due to a dissonance between the perceived and actual control over one’s risk for ovarian cancer [49]. In the present study, the mothers expressed challenges in relinquishing control over their roles, and this may serve as a barrier to attaining the psychological benefits of problem-focused coping. In a study of 89 individuals who were randomly assigned to either write about their feelings regarding a stressful life event or strategies to solve it, and who were then evaluated using a self-report questionnaire on their coping, integrating emotion processing was found to help the participants make better decisions in times of stress [27]. The present study finds that when both emotion-focused and problem-focused coping strategies are used, mothers can meaningfully increase their QoL as they define it because they are able to re-evaluate life and find balance. Thus, mothers with cancer should be encouraged to cultivate strategies within both styles of coping.

### 4.3. Sociocultural Context Surrounding the Relationship between Mothers’ Roles and Cancer

In the present study, the mothers’ self-image was impacted when cancer changed their caregiving capacity and perceived autonomy. Barnett and Hyde (2001) argue that one of the benefits of occupying multiple roles is improved self-image [23]. Thus, encouraging mothers to participate in multiple social roles may help them address the ways in which cancer can challenge their identities. However, it is critical to consider the social context that informs how mothers believe cancer should impact their role participation. Rottenberg (2014) suggests that social definitions of balance for women are ingrained within social hierarchies that equate productivity with value [50]. Moreover, the notion that one must strive to meet the demands they fulfilled as an able-bodied person reinforces the supercrip schema [36]. These social expectations were evident in the mothers’ guilt when they were unable to meet their role demands because of cancer, and particularly their parental roles; this is similar to a qualitative expressive-writing investigation of 27 Chinese-American breast cancer survivors that found that while mothers were struggling to fulfill their multiple roles, there was hesitancy to communicate their emotional needs and renegotiate the role demands within the family, which they accredited to Chinese cultural norms [51]. In the present study, the mothers cited their husbands’ employment demands as a key reason for their reluctancy to redistribute role demands to them. The notion that the amount of capital traded for labor signifies its merit or value produces a patriarchal hierarchy that insinuates that informal work, which is often performed by women, is less important [33]. The present study supports previous literature, but it also suggests that sociocultural conceptions of women’s labor and the expectations that disabled people occupy their roles according to able-bodied standards is pervasive in Canadian culture and can inform mothers’ experiences with cancer as it relates to managing role demands. It is plausible that mothers’ hesitancy to ask for support may result from internalized misogynistic and ableist views surrounding the value of their labor.

According to the present study’s findings, to achieve a fulfilling QoL, we suggest that mothers must be supported in deconstructing internalized sociocultural discourse. Moreover, the present study found that participating in hobbies was beneficial for the participants’ wellbeing, and thus mothers with cancer should be encouraged to cultivate self-care practices and hobbies, and to define themselves outside of their caregiving roles. Doing so may allow them to choose which roles they want to occupy, and to what extent those chosen roles should interact for them to live a personally fulfilling and balanced life.

The mothers articulated their perception of their family’s capacity to take on role demands and their opportunity to practice self-care; mothers with young children particularly had legitimate limitations on the help they could access. Financial security was cited by the mothers as a reason that prevented working spouses from taking on increased role demands typically addressed by the mother. Data from an American population-based survey determined that chemotherapy was significantly associated with financial worry and burden (e.g., failing to pay credit card bills, taking out loans, using savings, etc.) [52]. Even within publicly funded healthcare models, such as Canada’s, cancer patients are at an increased risk for financial strain [21,53]. As such, clinical guidance should pragmatically consider the nuanced intersection between sociocultural forces and a mother’s lived reality when making recommendations on how they should navigate their desirable and undesirable role demands during and after cancer treatment.

### 4.4. Implications and Future Directions

The RCMC model fills a gap in the current care for mothers with cancer. To the best of our knowledge, there are no clinical or research models that capture the roles that mothers with cancer occupy, their capacity to cope, and the sociocultural context that they operate within. The RCMC model synthesizes and translates the present study’s findings into a tool for mothers, researchers, and oncology teams.

Structuring the healthcare system such that mothers, and cancer patients in general, are viewed as experts on their own bodies, feel comfortable sharing how they are choosing to pursue health, and are confident that their oncology team is providing person-centered care can be better achieved by integrative care that situates the individual as the center of discourse and considers the “disabled experience” within a political and social context [35]. The RCMC model does this by considering the roles and multiple identities that surround the experiences of mothers with cancer, and the mothers expressed an interest in the RCMC model being used by healthcare providers. The model may be a useful guide for healthcare providers when initially discussing mothers’ cancer diagnoses, as it provides an easily understandable visual that highlights the interdependent relationships mothers occupy and supplies coping strategies to mitigate role conflict. Moreover, the RCMC model may be useful in informing further clinical research, as it highlights potentially useful outcome measures that are often not explored when evaluating clinical interventions. However, the RCMC model still needs to be evaluated in terms of its clinical utility. It is also unclear to what extent healthcare providers within oncology are familiar with the roles that mothers with cancer occupy, role expansion theory, and the sociocultural forces that shape how mothers navigate their role demands. Understanding healthcare providers’ perceptions on how these concepts relate would be a valuable avenue to ensure that oncology teams feel comfortable using the RCMC model and are confident that they can effectively engage in a discussion surrounding role coping for mothers with cancer.

The limitations of the present study include an insufficient sample size for generalizable quantitative results. Although the RCMC model reflects primarily qualitative results, its validity should be explored using a larger quantitative database. The lack of exploration into spouses’ perspectives on role coping is also a limitation, as cancer is a joint experience between couples, and relationship tension, decision making, and role demand distribution [54,55] must be continually addressed and negotiated. The sociocultural ring of the RCMC model also does not include the ethnic cultural context. There is evidence that cultural affiliation within the broader Western context impacts parental role functioning [34,51], and therefore may also impact mothers’ ability to cope with other roles. Additionally, no mothers in our sample had female spouses. Considering that much of the present study’s analysis was conducted through a socialist feminist lens, the coping strategies may be different when patriarchal labor distributions are less present within the relationship. Not including these identities limits the extent to which the RCMC model can be used to deconstruct the lived experiences of racialized and/or queer cancer patients.

## 5. Conclusions

The results of this mixed-methods study indicate that mothers continue to occupy multiple interacting roles during and after cancer treatment. Occupying these roles contributes to a good QoL when both emotion-focused and problem-focused coping strategies are used to set boundaries and find balance within, and between, roles. Mothers’ identities, and how they define successful role participation, is in part informed by sociocultural perceptions of labor for women and for individuals experiencing disability. The RCMC model synthesizes these findings into a potentially useful tool for patients and researchers, and the participants expressed a desire for the RCMC model to be adopted into clinical practice to facilitate more integrative and person-centered care:

“*I think what’s so great about just seeing this [the RCMC model] and it is that, to be able to be shown this as a discussion point on the start of this cancer journey, is it just helps you kind of prepare for it a little bit. I think that that’s why there’s like symbolic representation of all this, the study, and the contributions, and the thesis, and when you can distill it down into this, yes, it’s simplified, but to be able to look at this at the outset and think, well, these are all the things I’m actually going to have to think about. I think this is really powerful.*”(Lana)

## Figures and Tables

**Figure 1 cancers-15-01915-f001:**
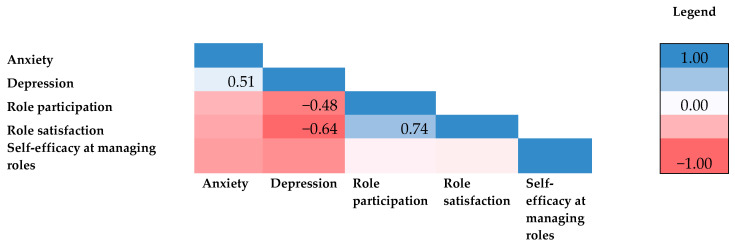
Heat map. Correlations between psychological and role functioning. *Note.* Pearson’s product–moment correlations were used to determine relationships between variables (*n* = 18).

**Figure 2 cancers-15-01915-f002:**
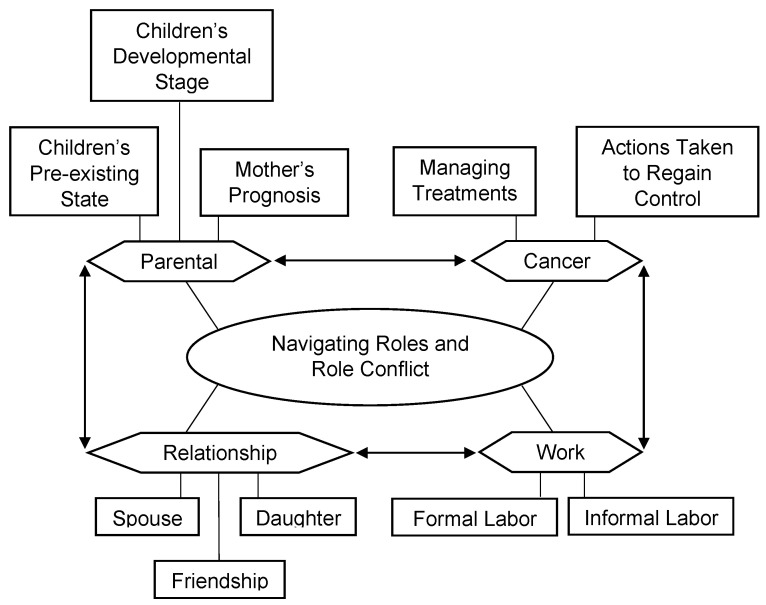
Visual map of Theme 1 by codes. *Note.* A line indicates that one component is a factor of another, and two arrows indicate a bidirectional influence.

**Figure 3 cancers-15-01915-f003:**
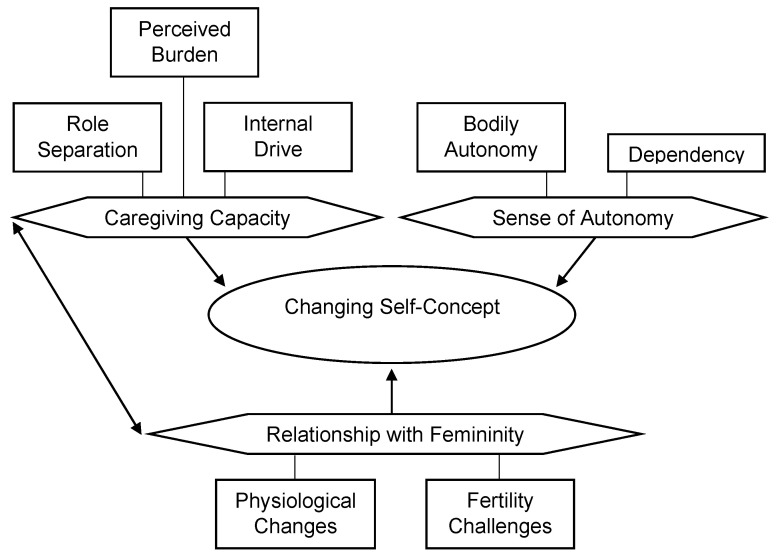
Visual map of Theme 2 by codes. *Note.* A line indicates that one component is a factor of another. One arrow indicates a unidirectional influence, and two arrows indicate a bidirectional influence.

**Figure 4 cancers-15-01915-f004:**
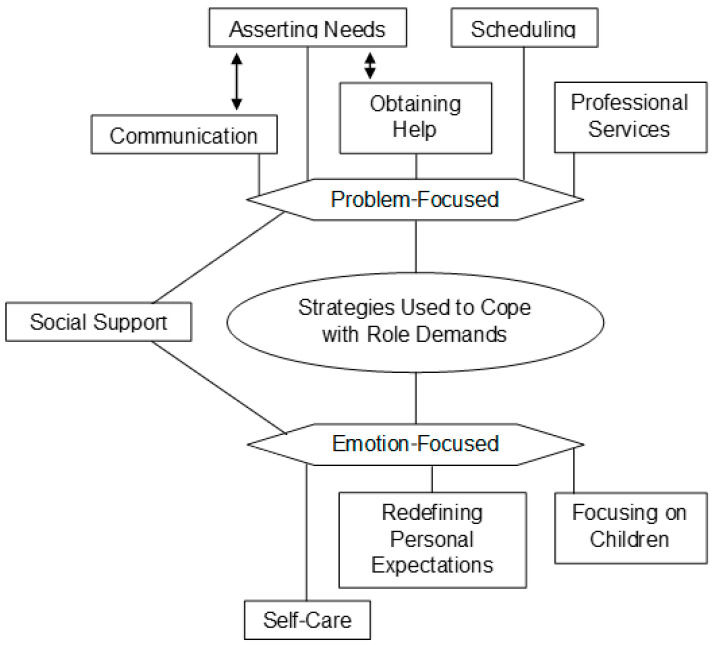
Visual map of Theme 3 by codes. *Note.* A line indicates that one component is a factor of another, and two arrows indicate a bidirectional influence.

**Figure 5 cancers-15-01915-f005:**
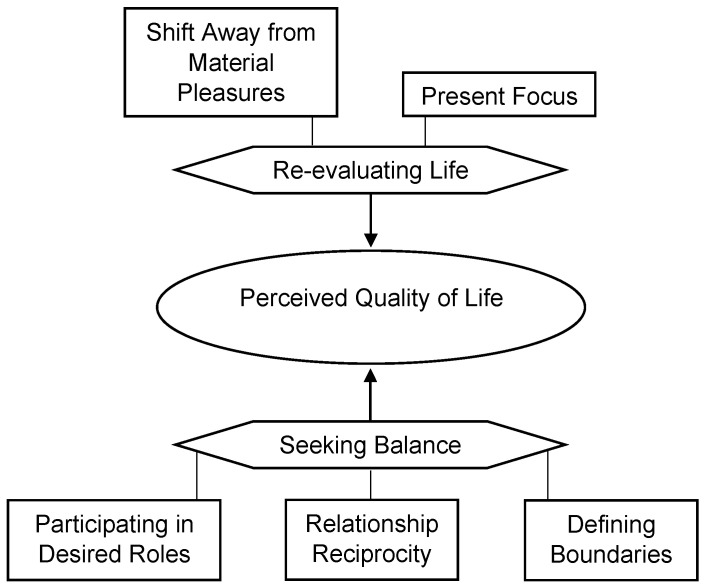
Visual map of Theme 4 by codes. *Note.* A line indicates that one component is a factor of another, and one arrow indicates a unidirectional influence.

**Figure 6 cancers-15-01915-f006:**
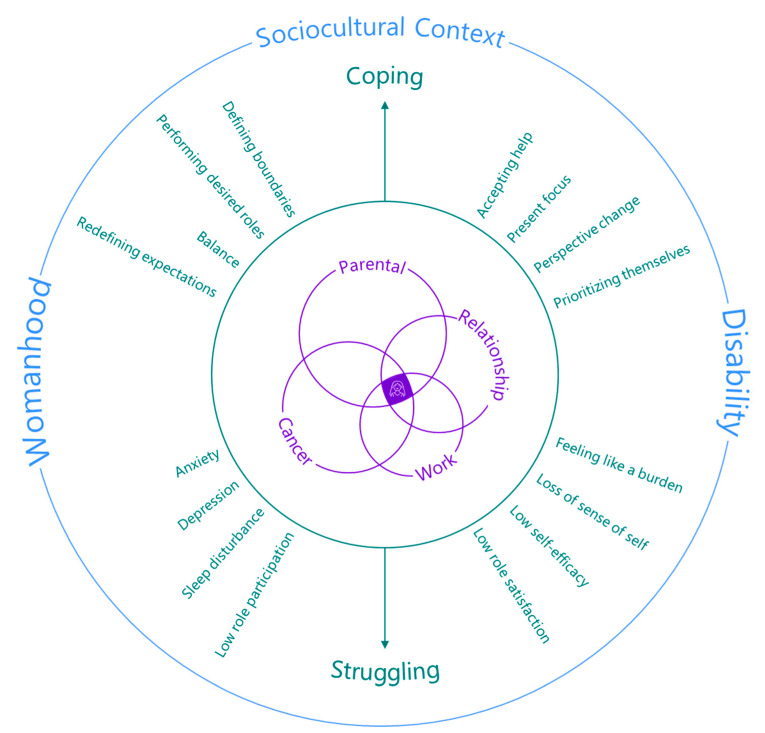
Role Coping as a Mother with Cancer (RCMC) model. *Note.* The RCMC model comprises three components: (1) the interacting roles that mothers with cancer occupy (the relative size of each circle signifies the priority that the mothers gave to the role) (purple); (2) the consequences of struggling to cope with role conflict caused by their multiple role demands and the strategies used to help them cope (green); (3) the sociocultural understanding of womanhood and disability that impacts how mothers cope with their roles and the expected demands within specific roles (blue). All the dimensions of the RCMC model are in a continual and multidirectional state of influence.

**Table 1 cancers-15-01915-t001:** Overview of interview transcript analysis.

Phase	Coding Method	Performed By
1. Researchers familiarize themselves with the data		A.S., volunteer 1, volunteer 2
2. Generating initial codes	Inductive approach	A.S., volunteer 1, volunteer 2
Iterative process	Consensus-based codebook	A.S., volunteer 1, volunteer 2
Review	Consensus-based codebook	A.S., volunteer 1, volunteer 2, J.D.
Data saturation	Final codebook	A.S., volunteer 1, volunteer 2
3. Searching for themes	Consensus approach based on emergent themes	A.S., volunteer 1, volunteer 2
4. Reviewing themes	Consensus approach based on literature	A.S., volunteer 1, volunteer 2
5. Defining themes	Consensus approach based on emergent themes and literature	A.S., volunteer 1, volunteer 2, J.D.,
6. Producing a report		A.S., volunteer 1, volunteer 2, J.D.

*Note.* Coding was conducted according to a priori and emergent content using a liberal coding standard. A priori codes were based on existing qualitative literature on mothers with cancer; however, approximately 70% of codes were emergent.

**Table 2 cancers-15-01915-t002:** Demographic, clinical, and health characteristics.

	*n*	%
Gender		
Female	18	100
Ethnicity *		
European	17	94
Central and South America	1	6
Japanese	1	6
Housing		
Urban	16	89
Rural	2	11
Highest level of education		
Community college	2	11
University (undergraduate)	11	61
University (professional/post-graduate)	5	28
Annual household income (CAD)		
From 51,000 to 80,000	3	17
From 81,000 to 120,000	1	6
>120,000	10	56
Prefer not to say	4	22
Primary source of income *		
Employment/business/investment	14	78
Pension/retirement	1	6
Social assistance	3	17
Family support	4	22
Employment status		
Full time	6	33
Part time	7	39
Full-time homemaker	4	22
Marital status		
Single	1	6
Married	12	67
Common-law	4	22
Separated	1	6
No. of dependent children		
One	9	50
Two	6	33
Three	3	17
Cancer type		
Breast	13	72
Hodgkin’s lymphoma	1	6
Non-Hodgkin’s lymphoma	2	11
Colorectal	1	6
Lung	1	6
Stage		
One	1	6
Two	6	33
Three	4	22
Four	5	28
Unsure	2	11
Treatments received *		
Chemotherapy	12	67
Radiation	11	61
Surgery	11	61
Hormone therapy	9	50
Immunotherapy	1	6
Targeted therapy	1	6
Stem-cell transplant	1	6
Prescription drug use		
Never	10	56
Rarely	6	53
Sometimes	1	6
Often	1	6
Marijuana use		
Never	14	78
Rarely	3	17
Often	1	6
Cigarette or tobacco use		
Never	17	94
Rarely	2	11
Alcohol use		
Never	8	44
Rarely	2	11
Sometimes	7	39
Often	1	6
Overindulging in foods		
Never	1	6
Rarely	9	50
Sometimes	4	22
Often	4	22
Weekly exercise		
<2 h	4	22
From 2 to 5 h	11	61
From 6 to 9 h	1	6
>9 h	2	11

* Participants could select multiple responses.

## Data Availability

The data presented in this study are available upon request from the corresponding author. The data are not publicly available due to the small sample size.

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
