# Peer review of "Mothers with Cancer: An Intersectional Mixed-Methods Study Investigating Role Demands and Perceived Coping Abilities"

_cancers, 2023, doi:10.3390/cancers15061915_

Round 1

Reviewer 1 Report

This study explores the coping strategies and role demands mother’s with cancer experience. The study utilized a mixed methods approach of validated patient-report questionnaires, interviews, and optional focus groups with a sample of 18 participants recruited across Canada using convenience and snowball sampling. The authors integrated the results from the questionnaires and interviews to form conclusions and to inform clinical care and future research. They also developed a model of Role Coping as a Mother of Cancer (RCMC).

The article expands understanding of the experience of cancer treatment and survivorship of mothers across multiple roles and responsibilities.  The authors use role expansion theory, intersectionality theory, socialist feminist theory and critical disability theory to describe the sociocultural context of personal experience and roles, healthcare relationship, and coping strategies.

The methods are clear and sample sizes for each of the 3 elements of the study are described.  The mixed method approach was limited by the small sample size and lower power of the quantitative component of the study.

The study limitations include small sample size for the quantitative component of the study and a relatively small sample size for the breadth and depth of the qualitative analyses especially given the exploration of coping related to work roles with only 6 participants working full-time.

1.    I enjoyed reading this article and learning about the authors’ thinking and conceptualization of the role challenges mothers with cancer may experience and the coping strategies employed.

2.      The term sociocultural context was introduced late in the paper.  It would be helpful to define or operationalize its meaning in the context of this paper earlier in the paper and to tie it to the theories being utilized in the mixed methods components of the study.

3.       Expand on the sample selection and any inclusions or exclusion criteria that were used. 

4.       Though all of the mothers have cancer, there were differences in the type of cancer, stage of cancer, and treatment modality. How might these difference impact coping?  

5.       Table 2 on page 5, confirm employment status of the sample or explain difference as total is 17 vs. 18.

6.       Visual maps for the identified qualitative themes, consider deleing the “one arrow” description note from maps where unidirectional influence is not indicated.

7.       Friendship Page 10 Line 357-363: This reader was perplexed by the shift from friendships to mother’s relationship

8.       Work role demands page 10, line 365-377 type of work, satisfaction with work prior to diagnosis, and financial implications of change are all factors that would influence this prioritization of this role.  Given that women varied in their FT, PT, or homemaker employment status discuss if there is true saturation for this role given the differences in these roles. Similarly, be more specific about what is intended as a confounding factor. Lastly, be cautious in inferring too much meaning to the lack of relationship between employment and psychological/role functioning reported from the quantitative portion of the study that has earlier in the manuscript been described as underpowered and used for descriptive purposes only.

9.       Emotion-focused coping, page 13, lines 480-482.  Social support is not listed on the visual map of theme 3 for problem-focused coping but is indicated as serving a dual purpose for both problem-focused and emotion-focused coping.  Clarification on what is meant and how social support is related to problem focused coping is needed.

10.   Page 14, lines 539-549. This section would benefit from revision to enhance clarify of the description. What is meant by “accessing professional services to offload role demands or alleviate their spousal relationships”?  Also line 545 and 546 I believe there is a typo here and perhaps the author means “financial capacity” rather than “financial toxicity”. And, again I urge caution with the comparison to the underpowered quantitative analysis.

11.   The discussion would benefit from revision to enhance clarity.  Consider using terms such as “similar to…” or “in contrast to…when discussing how the study findings are concordant or differ from findings of other studies.

12.   The developed model is intended to support clinical use – describe the ways in which it might do so. 

13.   Though all of the mothers have cancer, there were differences in the type of cancer, stage of cancer, and treatment modality. How might these difference impact coping?  

14.   Discuss if mothers are making realistic appraisals of their family’s capacity in regards to asking for help. Perhaps mothers don’t ask for help because they are appropriately assessing the level of support their children/spouse can manage to provide given their development, maturation, workload, and their coping capacity. Some of the quotes form participants touched on this. Implications for clinical support and guidance may vary should this be true.

Reviewer 2 Report

This is an interesting study. I do have some concerns that I have marked up on the paper. I think you over play the worth of the quantitative findings and relationships given the very small sample size. Methods section needs some further expansion as outlined on the manuscript. As a mixed methods study the approach to data integration is not articulated. 

There is also a caution here for me in the theoretical frameworks that fail to take account of the perspectives of partners and how cancer impacts them. While this isn't the focus of the study it should be captured in the limitations and recommendations for future work. I make some annotations on this on the manuscript. 
